# Efficient ultrafast field-driven spin current generation for spintronic terahertz frequency conversion

Igor Ilyakov[1] ✉, Arne Brataas [2], Thales V. A. G. de Oliveira [1], Alexey Ponomaryov[1], Jan-Christoph Deinert [1], Olav Hellwig [1,3], Jürgen Faßbender [1,4], Jürgen Lindner[1], Ruslan Salikhov [1] ✉ & Sergey Kovalev[1,5] ✉

Efficient generation and control of spin currents launched by terahertz (THz) radiation with subsequent ultrafast spin-to-charge conversion is the current challenge for the next generation of high-speed communication and data processing units. Here, we demonstrate that THz light can efficiently drive coherent angular momentum transfer in nanometer-thick ferromagnet/heavy-metal heterostructures. This process is non-resonant and does neither require external magnetic fields nor cryogenics. The efficiency of this process is more than one order of magnitude higher as compared to the recently observed THz-induced spin pumping in $MnF_2$ antiferromagnet. The coherently driven spin currents originate from the ultrafast spin Seebeck effect, caused by a THz-induced temperature imbalance in electronic and magnonic temperatures and fast relaxation of the electron-phonon system. Owing to the fact that the electron-phonon relaxation time is comparable with the period of a THz wave, the induced spin current results in THz second harmonic generation and THz optical rectification, providing a spintronic basis for THz frequency mixing and rectifying components.

The increasing demand for high-speed communication and data processing requires electronic components to operate in the terahertz (THz) frequency regime. Utilizing the spin degree of freedom is an attractive route to overcome limitations in terms of bandwidth and power consumption of currently used nanoscale electronic circuits[1,2]. The central challenge toward the realization of spin-based functionality is the search for an efficient generation and control of the electronic spin polarization on the picosecond (ps) timescale. A possible way to realize spin dynamics at THz frequencies is to exploit magnetic resonance in antiferromagnets[3,4] or ferrimagnets with large magnetic anisotropy[5,6]. The fast magnetization precession on ps-timescales in such materials results in spin currents (SCs) that may be

injected into an adjacent non-magnetic layer via adiabatic spin pumping[3,4]. This concept is efficiently and widely employed at gigahertz frequencies[1,7]. However, the first observation of THz-driven SCs in antiferromagnet/heavy-metal (HM) heterostructures has occurred only recently[3,4]. These studies demonstrate the detection of a down-converted dc-voltage component, resulting from the rectified SC contribution. As for the oscillating SC contribution, higher harmonic generation of the THz light has been predicted theoretically in heterostructures with spin-orbit coupling strength comparable to the exchange energy within a ferromagnet (FM)[8]. However, there has been no experimental evidence for spintronic THz-frequency up-conversion so far.

[1]Helmholtz-Zentrum Dresden-Rossendorf, Bautzner Landstr. 400, 01328 Dresden, Germany. [2]Center for Quantum Spintronics, Department of Physics, Norwegian University of Science and Technology, NO-7491 Trondheim, Norway. [3]Institute of Physics, Chemnitz University of Technology, 09107 Chemnitz, Germany. [4]Institute of Solid State and Materials Physics, Technische Universität Dresden, 01062 Dresden, Germany. [5]Technische Universität Dortmund, 44227 Dortmund, Germany. ✉e-mail: i.ilyakov@hzdr.de; r.salikhov@hzdr.de; sergey.kovalev@tu-dortmund.de

The light-induced ultrafast SC generation in FM/HM structures irradiated by near-infrared and extreme ultraviolet photons has received much attention in recent years[9–20]. In these studies, light-induced SCs in FM/HM heterostructures are employed to down-convert the light's high photon energy toward the THz frequency range. Femtosecond laser pulses excite electronic systems of the FM/HM structures, which leads to ultrafast SC flow from the FM to the HM layers. Afterward, this SC is transformed into a charge current via inverse spin-Hall effect (ISHE) in the HM layer emitting rectified electromagnetic pulses. However, whether a similar scenario can be observed under THz wave impact is still unknown.

In this work, we consider a strategy to generate ultrafast SCs triggered by THz electromagnetic radiation, analogous to the case of laser pulse excitation. The SCs are non-resonantly excited in FM/HM heterostructures. The coherent angular momentum transfer is attributed to the THz-induced ultrafast spin Seebeck effect (SSE), which was previously considered only in a non-coherent regime when using microwave or THz radiation[3,4]. In the case of laser pulse excitation[9–20], it has been demonstrated that the SC occurs almost simultaneously (within a few femtoseconds) with the photo-excited non-equilibrium electrons and decays exponentially with a sub-ps characteristic time of the electron-phonon energy transfer[16]. In the case of THz wave excitation, this characteristic time of electron-phonon scattering is, however, shorter than the period of the pumping wave oscillation, which, as we show here, leads to an ultrafast periodical modulation of the charge carrier temperature proportional to the THz wave instantaneous intensity, and results in coherent non-linear SCs excitation owing to the SSE. The coherent THz-induced spin transport is revealed by electro-optically detected radiation emission corresponding to THz second harmonic generation (TSHG) and THz optical rectification

(TOR) processes. The observed efficiency of the SC generation in our case is more than 40 times higher as compared to resonantly excited AFM/HM heterostructures[4] or 5 times higher compared with FM/HM heterostructures under laser light excitation[19,20].

## Results and discussion

We exploit time-resolved THz spectroscopy in Ta(3 nm)/Py(2 nm)/Pt(2 nm) (Py = $Ni_{81}Fe_{19}$, in the parentheses the corresponding thickness of each layer is given) metallic heterostructures, which represent efficient metallic spintronic emitters[10], or THz magnon generators[21]. The narrowband THz pulse with $\Omega$ = 0.5 THz central frequency (in the following referred to as fundamental beam), about 20% bandwidth, and 1.5 μJ pulse energy (see the "Methods" section for details) is focused onto the sample surface as schematically shown in Fig. 1a. We detect the TSHG and TOR signals in transmission geometry via electro-optical sampling (EOS) in the time-domain, using a ZnTe crystal and a synchronized femtosecond laser pulse. The near-field configuration is employed to detect both TSHG and TOR only in cross-polarization geometry due to the limited dynamic range of the measurements. At the same time, the background-free TSHG with high dynamic range and at different polarizations is studied in far-field configuration. The experimental schemes for far- and near-field geometries are presented in the Supplementary Information (Supplementary Fig. 1).

Figure 1a schematically depicts the underlying mechanism of the THz radiation up- and down-conversion. The linearly polarized (along the z-axis) electric field of a narrowband THz pulse propagates along the y-direction and is incident onto the sample from the Pt side. The light-induced electrical currents ($j_E$) oscillate in-plane within the whole metallic heterostructure at the fundamental frequency $\Omega$ = 0.5 THz. During the ps timescale of a half-cycle of the incoming THz wave, the

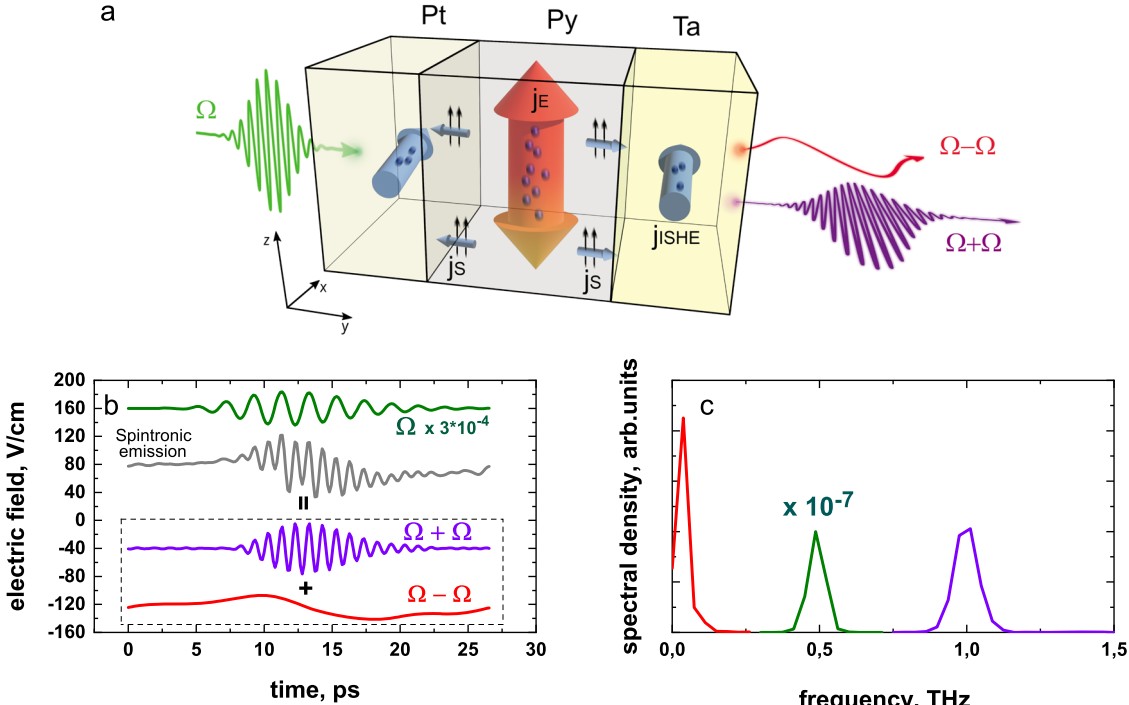

**Fig. 1 | Spintronic-based TSHG and TOR in a ferromagnet/heavy-metal heterostructure. a** Schematic representation of the THz conversion process. The THz-pump pulse with central frequency $\Omega$ (green curve) illuminates the Pt/Py/Ta sample, resulting in an electrical current $j_E$ in the whole heterostructure. The Py layer is magnetized along the z-axis (in plane). An ultrafast spin current, $j_s$ (blue arrows in the Py layer) is generated via the spin Seebeck effect and results in an electrical current $j_{ISHE}$ in the Pt and Ta layers (blue arrows) via the ISHE. The $j_{ISHE}$ in both heavy metals causes emission of THz radiation at each half of the period of the

incident THz wave resulting in second harmonic ($\Omega + \Omega$) and rectified zero frequency generation ($\Omega - \Omega$). **b** Time-domain and **c** corresponding FFT spectra of the near-field signals from the Pt(2 nm)/Py(2 nm)/Ta(3 nm) sample exposed to the 0.5 THz radiation. The green curve in (**b**) corresponds to the fundamental radiation ($\Omega$) and is scaled by $3 \times 10^{-4}$. The gray curve corresponds to the total measured near-field signal and contains both TSHG and TOR signals, which afterward are separated by implementing FFT filters (violet and red curves).

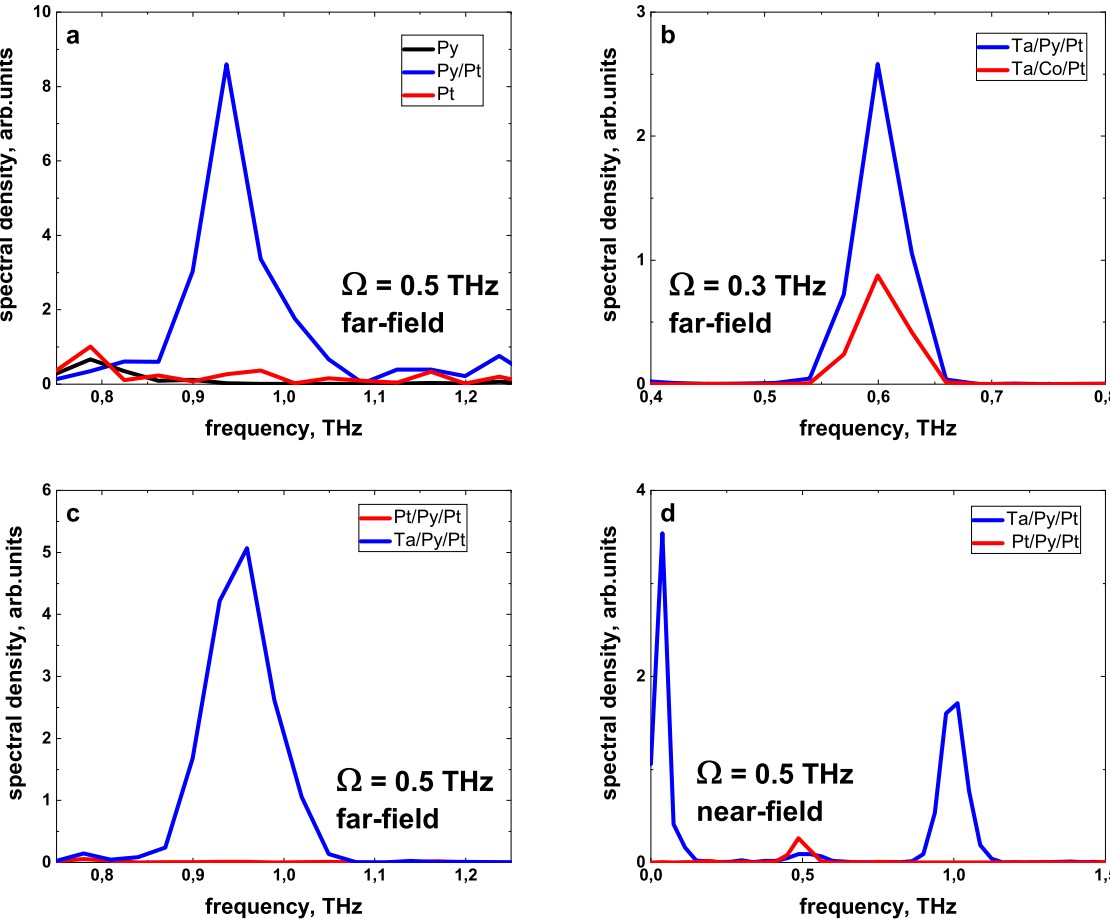

**Fig. 2 | Spintronic TSHG and TOR. a** Far-field emitted TSHG light spectrum from a single Py (black curve) and Pt (red curve) layer and from a Py/Pt (blue curve) bilayer at 0.5 THz pump frequency. The TSHG is evident only in the bilayer system. **b** TSHG spectrum measured in Py- and Co-based magnetic trilayers at 0.3 THz pump. **c** Comparison of the TSHG signals in two different samples with asymmetric (Ta(3 nm)/Py(2 nm)/Pt(2 nm)) and symmetric (Pt(2 nm)/Py(2 nm)/Pt(2 nm)) interfaces using 0.5 THz pump. In symmetric samples, the TSHG is inefficient. **d** Near-field TSHG and TOR signals from the same heterostructures at 0.5 THz pump,

illustrating that—similar to the TSHG—the TOR signal is absent in the symmetric samples. Both samples were magnetized vertically and the near-field signals had horizontal polarization that was orthogonal to the fundamental beam polarization (i.e., vertical). Note, the usage of the bandpass filters to suppress the fundamental radiation can lead to small frequency shifts of the TSHG toward lower values as seen in (**a**) and (**c**). This effect occurred because the maximum transmission for the nominally 1.0 THz BP filter was slightly off at about 0.95 THz.

electronic system of the heterostructure is excited, resulting in an almost instantaneous temperature rise. The effective temperatures of the phonon and magnon systems do not change significantly at these early times. This transient energy imbalance is similar to that of the ultrafast SSE[16,22–25], and results in ultrafast SCs ($j_s$), which propagate perpendicularly to the sample surface from the Py to the adjacent HM layers (i.e., along the $\pm y$-axis) with spin polarization **S** parallel to the sample magnetization. Note that this effect is different from the so-called "spin-dependent Seebeck effect" for which a gradient for the temperatures of the conduction electrons is required[26,27]. In the Pt and Ta layers, the spin current is converted into a transverse electrical current ($j_{ISHE} \sim [j_s \times S]$) via the inverse spin-Hall effect (ISHE). Finally, the ultrafast modulated $j_{ISHE}$ results in the generation of free space electromagnetic radiation, which is found to be polarized parallel to the $x$-axis. Since the spin-Hall angles in the Ta and Pt layers have opposite signs, the emitted electromagnetic waves at the two opposite interfaces interfere constructively, thus, amplifying the spintronic radiation amplitude. In the second half-cycle of the incident THz wave, where $j_E$ inverts its direction and the SC is independent of the $j_E$ direction, the process described above is repeated, thus resulting in electromagnetic radiation at the second harmonic frequency $\Omega + \Omega = 1$ THz. The TSHG amplitude scales quadratically with the THz pump field strength (at fundamental frequency) as we demonstrate in the

Supplementary Information (Supplementary Fig. 2). Besides the second harmonic, the zero-frequency mode of the rectified fundamental beam ($\Omega - \Omega$) is also detected in near-field geometry. In Fig. 1b we present the experimental signals detected in near-field configuration (TSHG and TOR) in comparison to the direct fundamental beam. The corresponding Fourier spectra for all signals are displayed in Fig. 1c.

In order to demonstrate that the TSHG is an interface effect between FM and HM layers, Fig. 2a presents the spectra from single Py and Pt layers, together with the spectrum obtained from a Py/Pt bilayer. No second harmonic signal is detected in a single 5 nm thin Py layer (black curve) as well as in 2 nm Pt (red curve). However, once we interface both layers within a Py(5 nm)/Pt(2 nm) stack, the TSHG signal becomes visible (blue curve in Fig. 2a). Furthermore, the TSHG is independent of the choice of the FM layer material, as we show in Fig. 2b. The Fourier spectra of the Ta(3 nm)/Py(2 nm)/Pt(2 nm) and Ta(3 nm)/Co(2 nm)/Pt(2 nm) samples demonstrate comparable TSHG intensities, indicating that the radiation efficiency is similar for both (Py and Co) ferromagnets. Finally, when we interface the Py layer symmetrically from both sides using either Pt or Ta layers, the TSHG signal vanishes, as it is shown in Fig. 2c for the Pt(2 nm)/Py(2 nm)/Pt(2 nm) sample. Additional data for a Ta(3 nm)/Py(9 nm)/Ta(3 nm) sample is presented in Supplementary Information, Supplementary Fig. 3. A similar scenario we observed in the near-field geometry, where

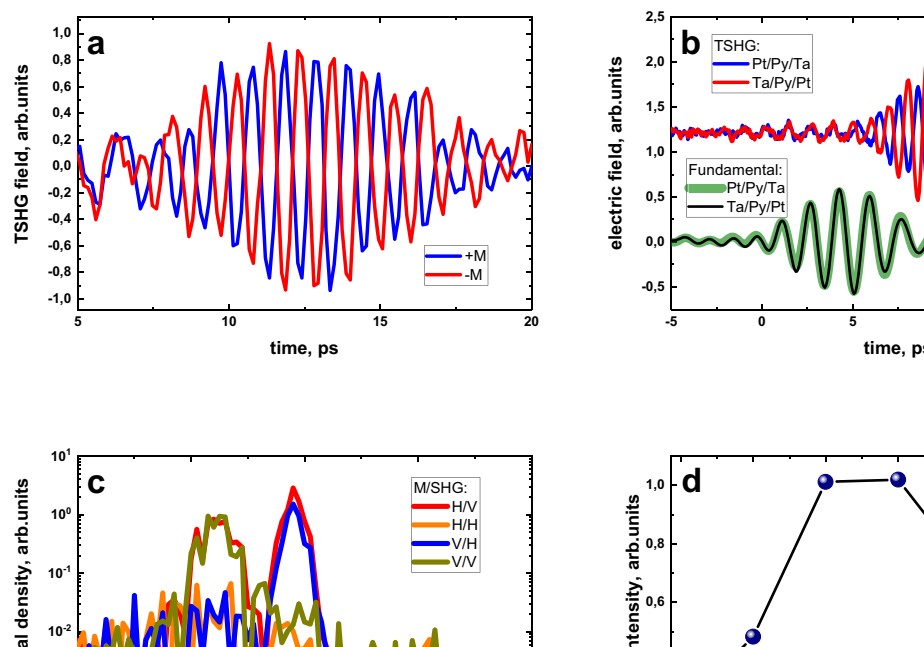

**Fig. 3 | Spintronic-based THz SHG symmetry. a** Time-domain measurements of TSHG in the Ta(3 nm)/Py(2 nm)/Pt(2 nm) sample in far-field geometry, when the sample is magnetized along two opposite directions. **b** Attenuated fundamental radiation and TSHG time-domain traces when the studied sample is facing the pump pulse with either Pt or Ta surfaces. **c** Spectral density of horizontal (TSHG-H) and vertical (TSHG-V) polarized SHG from Ta(3 nm)/Py(2 nm)/Pt(2 nm) hetero-structure, measured when the sample is magnetized horizontally (M-H) and verti-cally (M-V). The THz pump beam is polarized vertically. **d** Total SHG intensity as a function of pump beam ellipticity.

both TSHG and TOR signals show vanishing intensity for symmetric samples (Fig. 2d). Here, the generated $j_{ISHE}$ propagates along opposite directions, and, thus, the emitted THz radiation at the two interfaces is antiphase, thus leading to destructive interference of the electro-magnetic waves in the HM layers, consequently resulting in an almost zero signal.

In order to demonstrate that the TSHG is associated with an ultrafast SC, and thus confirming its origin (as described in Fig. 1a), in the following, we show that the TSHG polarization and phase resemble the geometry of spintronic THz emitters in metallic FM/HM heterostructures[9,10] as well as the longitudinal SSE configuration in metal/insulator systems[16]. Firstly, we demonstrate that the SC asso-ciated with the SSE ($j_s$ in Fig. 1a) is an odd function with respect to the direction of **M**. The EOS signal for opposite magnetization directions along the $z$-axis (see Fig. 1a) in the Ta(3 nm)/Py(2 nm)/Pt(2 nm) sample is presented in Fig. 3a. As evident from the figure, the phase of the TSHG shifts by 180°, when the sample is magnetized along opposite directions, indicating that $j_s$ is antisymmetric under the inversion of **M**. Furthermore, the magnetic field dependence of the TSHG amplitude coincides with the magnetization hysteresis curve, measured by vibrating sample magnetometry (Supplementary Information, Sup-plementary Fig. 4). In Fig. 3b, we present the far-field time-domain measurements of the TSHG, when the Pt/Py/Ta sample is illuminated from the Pt (blue curve) or Ta (red curve) side, respectively, i.e., when inverting the order of the Ta/Py/Pt layers with respect to the beam direction. Here, the sample is magnetized horizontally (along the $x$-axis in Fig. 1a) and the EOS is optimized for vertical polarization along the $z$-axis. Since the fundamental beam is vertically polarized as well, its leakage is visible with a lower period and at an earlier delay time (before 7 ps in Fig. 3b). The time delay between the leaking signal and

TSHG is caused by different effective refractive indices of the THz bandpass spectral filters at 0.5 and 1 THz. One sees that the phase of the fundamental beam at 0.5 THz does not change, whereas the TSHG phase inverts again by 180° when the sample is flipped. As a con-sequence, Fig. 3a, b demonstrates unambiguously that the second harmonic generation effect originates from interlayer spin transport.

To verify that the polarization of the TSHG signal is always orthogonal to the magnetization, we studied the dependence of the TSHG amplitude on the sample magnetization's orientation and emitted light polarization. For that, we measured the intensity of the vertically ($z$-axis) and horizontally ($x$-axis) polarized signals for differ-ent sample magnetization orientations within the film plane. The results are summarized in Fig. 3c. Here, the vertical (horizontal) polarization of the detected signal is labeled as SHG-V (H), and the sample's vertical (horizontal) magnetization is labeled as M-V (H). The TSHG, seen at 1 THz frequency, appears only in orthogonal config-urations with respect to the sample magnetization. When we measure the TSHG total intensity (sum of horizontal and vertical SHG intensity) as a function of the fundamental beam ellipticity (controlled by the orientation of the quarter-wave plate (QWP) for the incident THz beam), we find that the TSHG efficiency decreases with increasing the pump beam ellipticity as shown in Fig. 3d. At a QWP orientation of 0°, the fundamental beam is linearly polarized, at +45 and −45° the fun-damental beam becomes circularly polarized with opposite chirality (left or right). The 5-fold intensity drop for the circular polarization contradicts what has been reported for the SSE in FM/HM structures, where the optical light-induced THz emission is independent of the light ellipticity[16]. This implies that in our experiment the heating and cooling time of the electronic system is comparable with the half-cycle time of the fundamental wave. In the case of circular polarization, the

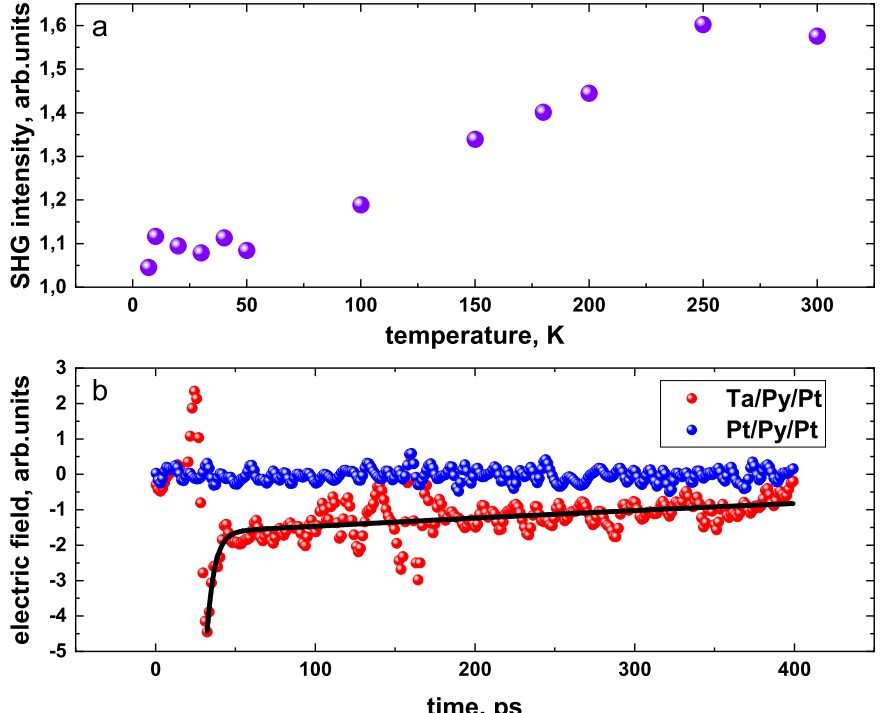

**Fig. 4 | Temperature dependence of TSHG and TOR signals. a** TSHG intensity as function of temperature for the Ta(3 nm)/Py(2 nm)/Pt(2 nm) sample. The second harmonic generation efficiency increases with temperature. **b** Comparison of the TOR amplitude for asymmetric (Ta(3 nm)/Py(2 nm)/Pt(2 nm)) and symmetric (Pt(2 nm)/Py(2 nm)/Pt(2 nm)) samples at extended time-delay scans (up to 400 ps).

While the symmetric sample exhibits zero amplitude signal, the asymmetric one shows THz-induced signals, which we approximate by a double exponential decay function (black solid line) with two different characteristic times. The longer characteristic time of about 400 ps is a measure of the phonon cooling relaxation time.

efficient sub-cycle electron heating and cooling modulations vanish as the THz field's absolute value is constant in time, leading to a reduced TSHG intensity. We note that similar processes were observed in studies of odd-harmonics generation in Dirac materials[28,29]. In the case of experiments with optical excitation, the pump-pulse duration is much shorter than the electron cooling time. Therefore, the light-induced energy cannot follow the optical pulse electric field dynamics and thus is independent of light polarization[16]. To conclude, the data presented in Fig. 3 confirms that the TSHG results from SC generation, while the origin of the THz-induced spin transport is given by ultrafast heating and cooling of the electronic system on a timescale shorter than (or comparable to) the half-cycle duration of the fundamental wave.

To gain further insight into the mechanisms involved in the TSHG, we study the temperature dependence of the TSHG intensity within a temperature range between 5 and 300 K (Fig. 4a). The TSHG efficiency increases with temperature. This behavior is in line with previous studies of the optically induced THz radiation efficiency[30,31] in magnetic heterostructures. The ISHE-generated current scales with the HM resistivity, resulting in proportionality between the TSHG amplitude and the squared resistivity of Pt (or Ta)[30]. The temperature dependence of the resistivity of the 4 nm Pt layer is shown in the Supplementary Information, Supplementary Fig. 8. To understand the THz-induced spin transfer dynamics at longer times, we measured TOR signals in near-field geometry within a 400 ps time window (Fig. 4b). The TOR signals exhibit two characteristic timescales at a few ps and a few hundred ps. Dynamics in the range of a few ps are determined by a combination of the THz pump pulse envelope heating and ultrafast electron-phonon thermalization, while dynamics in the range of a few hundred picoseconds are associated with phonon cooling[28,32]. This indicates that the TOR component of the SSE initially triggered by THz-induced electron heating, at later times transforms to phonon temperature relaxation. Both "fast" and "slow" TOR signals are odd with respect to the sample magnetization polarity and are absent in

symmetrically capped samples (Pt/Py/Pt). The optimal thicknesses of Py and Pt layers for TSHG correspond to about 2–3 nm (Supplementary Information, Supplementary Fig. 5). Such thicknesses are comparable with the characteristic spin diffusion length of the respective materials, and thus are similar for the optimal design of the spintronic THz emitters based on the same materials[11].

We demonstrate that TSHG as well as TOR signals originate from the spin transport induced by THz electromagnetic waves interacting with FM/HM heterostructures. The polarization of both signals is controlled by the magnetization direction in the FM layer. Furthermore, in the case of trilayers, TSHG and TOR signals can be detected only when the FM layer is sandwiched by HMs with opposite signs of the spin-Hall angle, or when the FM is capped with the HM from one side only, i.e., for bilayers. This implies that the SC is converted into electromagnetic radiation via the ISHE. The generated THz-wave spectral densities are more than $10^{-7}$ compared to the pump spectral density when about 150 kV/cm field strength of initial radiation is used. To compare THz-induced SC generation with other cases, we have calculated a second-order effective susceptibility $\chi_{eff}^{(2)}$ (see the Supplementary Information, Supplementary Fig. 6). This susceptibility can be used for comparison with spin pumping in AFM/HM structures and for comparison with laser-induced THz emission in FM/HM structures (in both these cases the generated electric field is also proportional to the pump pulse intensity). The observed efficiency of the SC generation ($\chi_{eff}^{(2)} = \frac{E_{2\Omega}}{E_{\Omega}^2} = 17.8 pm/V$, see the Supplementary Information, Supplementary Fig. 6) in our case is more than 40 times higher as compared to resonantly excited AFM/HM heterostructures ($\chi_{eff}^{(2)} = 0.4 pm/V$)[4] or 5 times higher compared with FM/HM heterostructures under laser light excitation ($\chi_{eff}^{(2)} = 3.8 pm/V$)[19,20]. Previous extensive studies of similar metallic heterostructures in the GHz and near-infrared frequency ranges did not report on spintronic second

harmonic generation. We attribute this to the relative timescales involved in the process. In the near-infrared regime, the field oscillation period is a few femtoseconds. Therefore, the electronic temperature cannot follow the light electromagnetic fields instantaneously, since the electron-electron scattering time is longer (of about 10 fs). In the GHz frequency range, the field period of the excitation is on the order of nanoseconds being much longer than the electron-phonon scattering time. Consequently, during the half-period of the electromagnetic stimuli, the electrons will have many scattering events with phonons, which strongly limits the induced changes in their temperature and decreases the SSE. In our study, the sub-ps timescale of electron-phonon relaxation is comparable to the temporal period of the THz wave. This leads to an efficient modulation of the electron temperature ($T_e$) at the second harmonic of the pump pulse frequency: at each half-period of the THz pulse, $T_e$ increases, while during the time when the THz E-field goes through zero, the $T_e$ releases its energy to the phonon system. The dynamics of electronic temperature will induce a spin voltage modulation, since in each transient state, the system aims to adapt the magnetization of the excited electronic state, and, therefore, the spin angular momentum of electrons is transferred to the adjacent HM layer. At the HM layer, spins are converted to charge due to ISHE, and THz radiation is released.

Having a qualitative explanation of the TSHG, for quantitative estimates, we consider the ultrafast SSE and the spin voltage accumulation at the interface as a possible microscopic origin of the SC. The maximum THz-pump peak power used in this work is about 58 MW/cm², which corresponds to about 100–200 K temperature modulation of the electron gas[18,32]. In metals, the heat capacity of the electrons is about one order of magnitude lower than the heat capacity of the phonons[18,32]. Thus in the SSE, the ultrafast SC is driven by the temperature difference $T_{diff}$ between magnons (in a ferromagnet) and electrons (in a heavy metal). The THz-induced $T_{diff}$ generates an electric field in the HM layer on the order of $E = \theta G_{mix} S T_{diff} / \sigma$, where $G_{mix}$ is the mixing conductance, $\theta$ is the spin-Hall angle, $S$ is the spin Seebeck coefficient, $\sigma$ is the conductivity. Using $G_{mix}S = 10^8 A/(m^2 K)$, the spin-Hall angle $\theta = 0.1$, and the typical conductivity in bulk Pt, $\sigma = 10^7 A/(Vm)$, we estimate that the induced electric field is of the order $E = 0.01$ V/cm per one Kelvin temperature difference[22,33]. At the same time, the measured TSHG field amplitude is about 39 V/cm (see Supplementary Fig. 6). To achieve such fields, would require a $T_{diff}$ of several 1000 K which is over an order of magnitude higher than the estimated THz-induced electron temperature modulation. The discrepancy between the observations and estimations could be caused by a reduction of the metal conductivity expected for the ultrathin film regime[34]. In ultrathin Pt films, the conductivity decreases by an order of magnitude compared to the bulk values[35,36]. We measured the resistivity of 2 and 10 nm thick Pt layers, and for the 2 nm layer it is about 30 times larger, which agrees well with previous work[35]. Additionally, we note that both HMs (Pt and Ta) radiate the TSHG, resulting in its multiplication. Both facts bring the experimentally observed TSHG efficiency into agreement with estimations based on the SSE. Moreover, at a delay time of hundreds of ps after the pump pulse, where excited electrons and phonons are at equilibrium, the spin transport is still present (Fig. 4b). Such long-term dynamics of the TOR signal support to identify the SSE as the origin of the spin transfer, where the SSE is initially triggered by an electron/magnon temperature imbalance and at later times driven by the increased temperature of the phonons.

We would like to note, that in the case of laser excitation, several other mechanisms have been considered for SC generation in FM/HM structures. In that case, electrons are excited from a d-band to highly mobile sp-like states with isotropic angular probability density of the velocity direction[37,38]. This leads to SC flow driven by super-diffusive transport of electrons excited in both FM and HM layers[37–40]. Another mechanism of spin current generation proposed in[41,42] considers a "band-type" transport, which occurs when initial and excited electron states in the FM have different band velocities, lifetimes, or energies. In this case, pump excitation can lead to an imbalance in terms of electron transport across the FM/HM interface, resulting in a spin-polarized electron flow. However, the case of THz excitation is quite different compared with the laser excitation because of much lower photon energy. In the case of THz excitation, the radiation energy is absorbed by the Drude mechanism, resulting in a completely different energy distribution of the electrons compared to laser excitation. Nevertheless, it is worth considering the possibility that ultrafast demagnetization[18], where the light-induced spin current is proportional to the time derivative of the magnetization[42,43], could also be a contributing mechanism. Therefore, to assess the possibility and determine the efficiency of the effects obtained with laser pumping for the case of THz excitation, additional investigations are required.

We have demonstrated the coherent non-linear SC excitation and frequency conversion in FM/HM heterostructures under THz-electromagnetic-wave impact. The observed radiation emission from this structure corresponds to non-resonant TSHG and TOR processes and shows that the efficiency of the SC generation is a few tens of times higher than in the case of the resonantly excited spin pumping in AFM/HM structures[4]. These processes are also more efficient than laser pulse optical rectification in similar structures used for THz emission[19,20]. The TSHG process also represents the first spintronic-based experimental demonstration of THz frequency multiplication. The observed non-linear field conversion efficiency is approximately $3 \times 10^{-4}$ (more than $10^{-7}$ by intensity), and TSHG and TOR signals have not reached any signatures of saturation at 58 MW/cm² of THz pump intensity. This implies that applying higher THz field strengths, for example by use of an additional metamaterial coverage, can significantly increase the TSHG and TOR conversion efficiency[44]. Another promising opportunity can be the use of 2D magnetic materials[45] with low electron heat capacity and based on multilayer heterostructures.

The observed SC generation and electromagnetic radiation emission are attributed to THz-induced ultrafast SSE and ISHE. We demonstrate that this process allows efficient coherent spin current excitation at ultrafast timescales. The THz SC excitation is demonstrated at room temperature, zero external magnetic fields, and does not require fulfilling magnetic resonant conditions. Moreover, these heterostructures can be fabricated with easy thin-film growth methods, wherein common 3d ferromagnets (i.e., Py, Co), and heavy metals are used (i.e., Pt, Ta). We believe that our results are important for paving the way toward the generation and control of THz-induced spin transfer dynamics, which will aid the design of next-generation high-speed spintronic devices.

## Methods

### Sample fabrication

All samples are fabricated at room temperature by dc-magnetron sputter deposition at 0.4 Pa Ar atmosphere in an ultrahigh-vacuum BESTEC system. We use 1-mm-thick double-side-polished Quartz ($SiO_2$) glass as a substrate. The sample homogeneity is ensured using the 30 rpm sample holder rotational stage during the deposition. The thickness for all metallic layers is controlled by deposition time. Prior to sample fabrication, the sputter rate of each individual material is calibrated using x-ray reflectivity (XRR) data of the corresponding film material.

### Experimental setup

To generate high-field THz radiation, we used both laser-based and accelerator-based sources. The laser-based THz source was based on a tilted pulse front generation (TPFG) scheme using 9 mJ, 35 fs laser pulses at 800 nm central wavelength and at 1 kHz repetition rate. The

TPFG THz pulses are initially broadband with frequency content spanning from 200 to 1.5 THz. To convert them to narrowband radiation, two 500 GHz narrowband filters with 20% bandwidth were used. Focused on the sample, the THz pulse after these filters had about 1.5 μJ pulse energy and 800 μm diameter (FWHM), resulting in 148 kV/cm peak field strength. The THz radiation transmitted through the sample was characterized via electro-optical sampling in a 2 mm < 100 > ZnTe crystal using about 1% splitting from the main (9 mJ) laser pulse. To suppress the strong fundamental radiation at 0.5 THz, additional THz bandpass filters with high transmission at the SHG frequency are placed right after the sample. To measure both THz SHG and TOR signals we used a near-field geometry (see the Supplementary Information). To study the temperature dependence of the THz SHG, we used the accelerator-based THz source TELBE located at Helmholtz-Zentrum Dresden-Rossendorf. This source was tuned to 300 GHz central wavelength and was synchronized to an external 30 fs laser system used for EOS[46]. We use a permanent magnet, which is mounted above the samples, and can be rotated about the sample holder to adjust the magnetization direction within the sample plane. The magnet delivers about 50 mT magnetic field at the sample position, which is fully sufficient for the in-plane magnetic saturation of Py films. The magnet with 100 mT in-plane magnetic field was used for Co film.

## Data availability

The data that support the findings of this study are available from the Supplementary Information or at https://doi.org/10.14278/rodare.2516.

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

## Acknowledgements

Parts of this research were carried out at ELBE at the Helmholtz-Zentrum Dresden-Rossendorf e.V., a member of the Helmholtz Association. The Research Council of Norway (RCN) supported A.B. through its Centers of Excellence funding scheme, project 262633, "QuSpin".

## Author contributions

S.K., R.S., I.I. and J.L. designed the project. I.I. and S.K. conceived the idea of EOS measurements of THz-induced spin transfer and developed optical setups. R.S. and O.H. designed and prepared samples. S.K., I.I. and A.P. performed the laser-based measurements. S.K., I.I., R.S., A.P., T.O. and J.D. performed the measurements using the TELBE source. A.B. and S.K. performed the estimations of the spin transfer-based THz SHG. S.K., R.S., I.I., J.L. and A.B. wrote the manuscript with contributions from all authors. I.I., S.K., R.S. and A.B. revised the manuscript with contributions from all authors.

## Funding

## Competing interests

The authors declare no competing interests.
