## [Peer Review File · Nature Communications]

Reviewers' Comments:

Reviewer #1:

Remarks to the Author:

The manuscript reports on non-linear THz responses of Ta/py/Pt metallic structures. A second harmonic emission and a rectified emission are observed in response to a THz excitation. The experimental observations are discussed in terms of spin Seebeck effect, that would be responsible for the creation of a spin current and of inverse spin Hall effect, that would provide a spin to charge conversion mechanism in the heavy metals responsible for a THz emission.

The experimental results presented are very interesting. However, I have some concerns on the interpretation of the results and in particular on the microscopic mechanisms invoked.

1) The generation of spin currents is discussed only on the basis of the spin Seebeck effect. However, the validity of this mechanism is not sufficiently justified in the current manuscript. The penetration depth of the THz field exceeds the thickness of the Pt(2nm)/py(2nm)/Ta(3nm) structure. Therefore, the existence of a temperature gradient along the longitudinal direction is not a priori established. A thermal gradient in the light propagation direction is a necessary condition in order for the emission to take place, within the scenario proposed in the paper.

2) Another possible mechanism, namely ultrafast demagnetization is not adequately addressed. For example, in Chekhov et al. Phys. Rev. X 11, 041055 (2021) a THz-induced demagnetization of ferromagnetic metals is shown to exist and to be very similar to the one observed for optical pump at 400 nm. This ultrafast demagnetization process would be an alternative non-local mechanism for injection of angular momentum in the heavy metal layers.

The other mechanism proposed, namely the inverse spin Hall effect is appropriately addressed through measurements symmetry properties of the emission (figure 3) and its temperature dependence (figure 4). A similar grounding is needed also to support the spin current generation mechanism.

Reviewer #2:

Remarks to the Author:

Review of Nature Communications NCOMMS-23-01825-T

The Manuscript describes the use of Ta/Py/Pt heterostructures for the purposes of THz frequency conversion. After reading through the Manuscript and Supplementary Information a number of times, unfortunately, I cannot recommend publication of the Manuscript in NC in its present form. I would recommend that the main message be brought forward in a quantitative way, all arguments distilled to their essence. The following is a list of points that the Authors may wish to consider in their future work on the Manuscript:

1. When claims of efficiencies improvements are made, as in the Abstract, they have to be substantiated with actual numbers for what has been achieved and what has been the previous best. Perhaps in spin current density per unit of incident power per unit bandwidth.
2. Other non-linear components can be used, at least in the 0.1 – 0.2 THz range (e.g. Schottky diodes), for harmonic generation and parametric rectification. The question arises as to how the spintronic structures compare with respect to the semiconducting ones.
3. THz frequency upconversion using spintronic structures could be potentially interesting for detection, if potentially matched to available IR detection technology. What harmonics are expected to be generated and with what relative intensity?
4. An order of magnitude is claimed in spin current generation, when compared to AFM/HM structures or FM/HM structures, but only one reference (to the first case only) and no numbers are provided.
5. From figure 1, it would appear that the ratio of spectral density between the carrier and the second harmonic, and the carrier and the rectified component is bigger than 10^7 . This appears quite high, in view of the claimed efficiencies and requires a more detailed explanation.
6. The Authors make the argument that the TSHG and TOR signals are detected only when the FM

layer is sandwiched between layers of HM of opposite spin-Hall sign. The question arises if evidence can be provided in tri-layers rather than using bilayers?

7. On Page 8 an explicit comment is made on the spectral densities of the generated THz waves, at electric field magnitude of 150 kV/cm. As the effects would be nonlinear both in the optical and THz regions, it begs the question as to what is the fair way of comparison – by intensity (squared amplitudes) or directly by intensity?

8. On the same page, bulk Pt conductivity is involved in the argument, only to be alleged for the discrepancy of TSHG field amplitude. It is quite clear that layers of such thicknesses cannot remotely reach bulk conductivities and a more refined estimates of conductivity may be needed (from optical complex refractive index measurements or perhaps microwave reflection/absorption measurements).

9. The argument of Note 1 of the SI is very schematic. It appears that there is some dimensional factor – likely an elementary charge missing from equation 8.

10. If the proportionality in Note 1 is based on the schematic square root, quasi-free electron gas dispersion in 3D, then it is probably better to have the final Js to be expressed as a function of the Fermi level spin polarization and not be limited by an assumption of $\Delta = E_f / 2$.

11. Also in Note 1, the temporal aspect is lost after eq. 2. What assumptions are needed to justify this. The dimensionless factor R is not defined in eq. 1. And the Gamma factors for electron-spin and electron-phonon coupling cannot possibly have dimension of time as of the way they enter the exponents.

We appreciate the opportunity to resubmit the manuscript and thank the reviewers for their time and valuable comments. Below, we provide a point-by-point response to the reviewers' comments. We have also modified the manuscript's text (new text is highlighted in red) based on the reviewers' requests. We hope that our response and the manuscript modifications address all the points raised by the reviewers and that our manuscript now meets the standards of Nature Communications.

Reviewer #1: The manuscript reports on non-linear THz responses of Ta/Py/Pt metallic structures. A second harmonic emission and a rectified emission are observed in response to a THz excitation. The experimental observations are discussed in terms of spin Seebeck effect, that would be responsible for the creation of a spin current and of inverse spin Hall effect, that would provide a spin to charge conversion mechanism in the heavy metals responsible for a THz emission. The experimental results presented are very interesting. However, I have some concerns on the interpretation of the results and in particular on the microscopic mechanisms invoked.

- 1) The generation of spin currents is discussed only on the basis of the spin Seebeck effect. However, the validity of this mechanism is not sufficiently justified in the current manuscript. The penetration depth of the THz field exceeds the thickness of the Pt(2nm)/Py(2nm)/Ta(3nm) structure. Therefore, the existence of a temperature gradient along the longitudinal direction is not a priori established. A thermal gradient in the light propagation direction is a necessary condition in order for the emission to take place, within the scenario proposed in the paper.

Reply: We would like to thank the reviewer for bringing up this question. We agree that the sample thickness is well below the THz-field penetration depth. In the case of the spin Seebeck effect, a spatial gradient of temperature is indeed required to initiate a spin current flow. However, in our case we do not consider temperature gradients within the conduction electron system, as is discussed in the context of the “spin-dependent Seebeck effect” [Johnson M and Silsbee R H 1987 *Phys. Rev. B* **35** 4959–72; 2006 *Concepts in Spin Electronics* ed S Maekawa (Oxford: Oxford University Press)], but rather differences of the temperatures between electrons in the heavy metal (HM) layer and magnons in the ferromagnetic layer (FM) layer [Annu. Rev. Condens. Matter Phys. 2023. **14**:129–51; Hiroto Adachi *et al*, 2013 *Rep. Prog. Phys.* **76** 036501; Jaworski C M *et al* 2010 *Nature Mater.* **9** 898; Uchida K *et al* 2010 *Nature Mater.* **9** 894–7; *Phys. Rev. B* **81**, 214418 (2010)]. In the manuscript, we thus consider the “ultrafast” spin Seebeck effect, which is activated only in the limited time interval when electrons in the HM and magnons in the FM have different temperatures. During excitation, the whole metallic sample is heated homogeneously, however the electron and magnon systems exhibit different absolute temperatures, which vary in time. This happens because the THz pump first deposits its energy on the electronic systems of the HM and FM layers by Drude absorption. After that, this energy is redistributed to phonons and magnons (in the case of the FM layer). This temperature difference between the electron system of the HM and magnon system of the FM serves as a source of spin current flow during the time window when the electron and magnon systems are not yet in equilibrium. To better introduce this mechanism in the manuscript, we have added the following sentences on page 5:

“Note, this effect is different from the so called “spin-dependent Seebeck effect” for which a gradient for the temperatures of the conduction electrons is required^{26,27}.”

- 2) Another possible mechanism, namely ultrafast demagnetization is not adequately addressed. For example, in Chekhov *et al*. *Phys. Rev. X* **11**, 041055 (2021), a THz-induced demagnetization of ferromagnetic metals exists and is very similar to the one observed for optical pump at 400 nm. This ultrafast demagnetization process would be an alternative non-local mechanism

for injection of angular momentum in the heavy metal layers. The other mechanism proposed, namely the inverse spin Hall effect is appropriately addressed through measurements symmetry properties of the emission (figure 3) and its temperature dependence (figure 4). A similar grounding is needed also to support the spin current generation mechanism.

Reply: We thank the reviewer for this valuable comment. Below, we discuss other potential mechanisms of spin current generation.

Ultrafast demagnetization, a decrease of the net magnetization, in spintronic heterostructures has been widely studied in the case of optical laser pumping. In this case, electrons are excited from a d-band to highly mobile sp-like states with isotropic angular probability density of the velocity direction [PRL 105, 027203 (2010)]. This leads to ultrafast demagnetization through spin-flip scattering and superdiffusive spin transport to a neighboring layer [Phys. Rev. Lett. 119, 107203 (2017); <https://arxiv.org/ftp/arxiv/papers/2211/2211.15135.pdf>]. In particular, this "spin transport" channel of demagnetization in heterostructures is driven by superdiffusive transport of electrons excited in both FM and HM layers [PRL 105, 027203 (2010); Phys. Rev. Lett. 122, 067202 (2019); <https://arxiv.org/ftp/arxiv/papers/2211/2211.15135.pdf>]. In the case of THz excitation, the radiation energy is absorbed by the Drude mechanism, resulting in a completely different energy distribution of the electrons compared to laser excitation (photon energy of the THz radiation we used is 2 meV which is much lower than the thermal energy – 300 K corresponds to about 25 meV). Therefore, the previous results obtained with laser pumping cannot be directly transferred to THz excitation. Compared to the long history of ultrafast spin current excitation by laser pulses, the case of THz wave excitation is much less investigated. To assess the possibility of a superdiffusive spin transport effect in the case of THz excitation, additional investigations are required.

In recent work by Chekhov et al. [Phys. Rev. X 11, 041055 (2021)], the authors study the demagnetization of an iron film by laser (3.1 eV) and THz (4.1 meV) pulses. They show that the spin-flip scattering process leads to demagnetization of the sample in a similar way for laser and THz excitations. However, in that paper the authors do not consider any scenarios of excitation of spin transport between FM and HM layers.

Another mechanism of spin current generation proposed in [Phys. Rev. B 106, 144427 (2022)] considers a "band-type" transport, which occurs when initial and excited electron states in the FM have different band velocities, lifetimes, or energies. In this case, pump excitation can lead to an imbalance in terms of electron transport across the FM/HM interface, resulting in a spin-polarized electron flow. This mechanism is considered only for optical excitation, which provides a necessary difference in the parameters of the initial and excited electronic states due to the large photon energy. It is also worth noting that the presented analysis does not allow for a quantitative estimation of the spin current amplitudes due to unmeasurable parameters such as the microscopic parameters of the Stoner model.

The mechanism based on the spin-Seebeck effect allows a quantitative estimation of the THz-induced spin-transport process and shows good agreement with the experimental results. Further experimental and theoretical work is needed to understand in more details the THz field induced spin current generation on a microscopic level, in particular the interplay between ultrafast demagnetization, spin- and spin-dependent Seebeck processes.

Page 9:

The sentence

"Besides the SSE, the recently discussed pyrospintronic effect (PSE) in metallic heterostructures may also be responsible for inducing TSHG and TOR via modulation of the spin potential μ_0 . Here, the increased temperature results in an excess of the spin density within the FM layer that triggers a spin current flow from the FM to the HM layer³⁴. However, in the PSE, $\mu_0 \propto T_0 \Delta T_e$ ³⁷, where T_0 is the

temperature of the Fermi-Dirac distribution of electrons before the laser excitation, and ΔT_e is the light-induced increase of the electron temperature. Therefore, the PSE efficiency should be proportional to the sample temperature (see also the Supplementary Information, Note 1). As we see in Fig. 4a, the observed TSHG only slightly depends on the sample temperature, and changes by 60% in power from 10 K to 300 K.”

Was replaced with

“We would like to note, that in the case of laser excitation several other mechanisms have been considered for SC generation in FM/HM structures. In that case, electrons are excited from a d-band to highly mobile sp-like states with isotropic angular probability density of the velocity direction^{37,38}. This leads to SC flow driven by superdiffusive transport of electrons excited in both FM and HM layers³⁷⁻⁴⁰. Another mechanism of spin current generation proposed in^{41,42} considers a “band-type” transport, which occurs when initial and excited electron states in the FM have different band velocities, lifetimes, or energies. In this case, pump excitation can lead to an imbalance in terms of electron transport across the FM/HM interface, resulting in a spin-polarized electron flow. However, the case of THz excitation is quite different comparing with the laser excitation because of much lower photon energy. In the case of THz excitation, the radiation energy is absorbed by the Drude mechanism, resulting in a completely different energy distribution of the electrons compared to laser excitation. Nevertheless, it is worth considering the possibility that ultrafast demagnetization¹⁸, where the light-induced spin current is proportional to the time derivative of the magnetization^{42,43}, could also be a contributing mechanism. Therefore, to assess the possibility and determine efficiency of the effects obtained with laser pumping for the case of THz excitation, additional investigations are required.”

We greatly appreciate the reviewer's positive attitude toward our work and valuable feedback on interpreting our results. We hope that our response and the modifications of the manuscript address all the points raised by the reviewer.

Reviewer #2: The Manuscript describes the use of Ta/Py/Pt heterostructures for the purposes of THz frequency conversion. After reading through the Manuscript and Supplementary Information a number of times, unfortunately, I cannot recommend publication of the Manuscript in NC in its present form. I would recommend that the main message be brought forward in a quantitative way, all arguments distilled to their essence. The following is a list of points that the Authors may wish to consider in their future work on the Manuscript:

We would like to thank the reviewer for valuable comments. In response to them, we have made a number of changes to the text. Please find below our point-by-point response.

We would also like to point out that the spintronic THz frequency conversion is not the only novelty of the manuscript. The manuscript presents first results on THz induced nonlinear ultrafast spin transfer currents in Ta/Py/Pt heterostructures, which were measured by THz frequency conversion. Compared to the recently published data on THz induced spin-pumping in MnF₂, we observe that the efficiency of spin current excitation is more than 40 times higher in the case of Ta/Py/Pt structure.

1. When claims of efficiencies improvements are made, as in the Abstract, they have to be substantiated with actual numbers for what has been achieved and what has been the previous best. Perhaps in spin current density per unit of incident power per unit bandwidth.

Reply: The spin-pumping and spin-Seebeck effects generate spin currents that are proportional to the intensity of the driving electromagnetic wave. These spin currents propagate to a neighboring heavy metal layer (often platinum) where they are converted to electric currents by the Inverse Spin Hall Effect (ISHE). This ISHE current/voltage can be measured with a voltmeter. Therefore, spin-pumping and spin-Seebeck effects can be characterized by the effective second order susceptibility $\chi_{eff}^{(2)}$, which is determined as the ratio of $E_{2\Omega, \Omega-\Omega}/E_{\Omega}^2$, where $E_{2\Omega, \Omega-\Omega}$ is the generated up-converted electric field at the frequency of 2Ω or down-converted rectified electric field at the frequency of $\Omega-\Omega$, E_{Ω}^2 is the intensity of the pumping THz wave at the frequency of Ω . We have implemented the $\chi_{eff}^{(2)}$ in the text and added this explanation and actual numbers on Page 8:

The previous sentence, “Such efficiency is more than one order of magnitude higher than the efficiency of spintronic THz emitters under optical excitation^{29,38}, or resonant THz spin-pumping in an MnF₂ antiferromagnet⁴ (Supplementary Information, Extended Figure 6).”

Was replaced with

“To compare THz-induced SC generation with other cases, we have calculated a second order effective susceptibility $\chi_{eff}^{(2)}$ (see the Supplementary Information, Extended Figure 6). This susceptibility can be used for comparison with spin pumping in AFM/HM structures and for comparison with laser-induced THz emission in FM/HM structures (in both these cases the generated electric field is also proportional to the pump pulse intensity). The observed efficiency of the SC generation ($\chi_{eff}^{(2)} = \frac{E_{2\Omega}}{E_{\Omega}^2} = 17.8 \text{ pm/V}$, see the Supplementary Information, Extended Figure 6) in our case is more than 40 times higher as compared to resonantly excited AFM/HM heterostructures ($\chi_{eff}^{(2)} = 0.4 \text{ pm/V}$)⁴ or 5 times higher comparing with FM/HM heterostructures under laser light excitation ($\chi_{eff}^{(2)} = 3.8 \text{ pm/V}$)^{19,20}.”

On a page 2:

The initial sentence, “Most importantly, the observed efficiency of the SC generation in this case is ten times higher as compared to resonantly excited AFM/HM heterostructures⁴ or in FM/HM heterostructures under laser light excitation.” was replaced with “The observed

efficiency of the SC generation in our case is more than 40 times higher as compared to resonantly excited AFM/HM heterostructures⁴ or 5 times higher comparing with FM/HM heterostructures under laser light excitation.^{19,20}

2. Other non-linear components can be used, at least in the 0.1 – 0.2 THz range (e.g. Schottky diodes), for harmonic generation and parametric rectification. The question arises as to how the spintronic structures compare with respect to the semiconducting ones.

Reply: We agree with the reviewer that other currently more efficient frequency conversion devices, such as Schottky diodes, are available for the sub-THz frequency range. However, these devices operate via physical principles not related to spin transfer. Our manuscript demonstrates that THz pulses can induce ultrafast nonlinear spin transport, which is converted by the ISHE into electromagnetic waves related to the TSHG and TOR signals. The demonstrated spintronic nonlinear frequency conversion may be of particular interest for technologies based on spin transport dynamics. We have compared the spintronic TSHG and TOR with another mechanism of THz light induced coherent spin transport dynamics, namely spin pumping, where the spin current injection is also determined by the ISHE in the Pt layer and have shown that the efficiency of spin current excitation is 40 times higher in our case.

In addition, we note that electronic-based devices have been optimized for several decades, while the proposed spintronic frequency multiplication is demonstrated for the first time. Therefore, there is lot of room for improvement, e.g., by using metamaterials for resonant THz absorption and field enhancement, or recently emerging 2D magnetic materials with low electron heat capacity and based on multilayer heterostructures, which needs to be explored in follow-up research.

We discuss this point in the conclusion:

“The observed non-linear field conversion efficiency is approximately 3×10^{-4} (more than 10^{-7} by intensity), and TSHG and TOR signals have not reached any signatures of saturation at 58 MW/cm² of THz pump intensity. This implies that applying higher THz field strengths, for example by use of an additional metamaterial coverage, can significantly increase the TSHG and TOR conversion efficiency³⁵. Another promising opportunity can be the use of 2D magnetic materials⁴⁵ with low electron heat capacity and based on multilayer heterostructures.”

3. THz frequency upconversion using spintronic structures could be potentially interesting for detection, if potentially matched to available IR detection technology. What harmonics are expected to be generated and with what relative intensity?

Reply: Spintronic high harmonic generation may be very interesting for potential applications, but it was beyond the scope of our research. In the manuscript, we have demonstrated the spintronic second harmonic generation and rectification process, and at the moment we find it difficult to predict the generation of higher order harmonics.

4. An order of magnitude is claimed in spin current generation, when compared to AFM/HM structures or FM/HM structures, but only one reference (to the first case only) and no numbers are provided.

Reply: To better introduce this point, we have added a second order effective susceptibility $\chi_{eff}^{(2)}$ to the main text, and its calculations into the Supplementary Information. This susceptibility can be used for comparison with spin pumping in AFM/HM structures (see also a reply to the first comment) and for comparison with laser-induced THz emission in FM/HM structures (in this case the generated electric field is also proportional to the pump intensity).

To compare the $\chi_{eff}^{(2)}$ with the case of laser excitation, we refer to the recent work [38] that can be used to estimate the absolute values of the optical-THz conversion efficiency. In this case the $\chi_{eff}^{(2)}$ is 3.8 pm/V, which is about 5 times lower than in the case of THz excitation (17.8 pm/V).

There are indeed many papers on the STE optimization and enhancement using laser pulse excitation. But in most of the papers there is not enough information to calculate absolute conversion of STE. In recent paper [38], the authors managed to generate few nJ THz pulses using large area STE under mJ laser pulse excitation. For this case, it is possible to estimate the absolute efficiency and corresponding second order susceptibility of STE.

Regarding the articles on spin pumping in THz light-triggered AFM/HM structures, this effect has been observed experimentally for the first time only recently [3-4]. In one of these papers [3], the authors used a specially designed contact structure to enhance ISHE. Due to its small size, this structure can additionally enhance the incident THz wave field, which prevents a direct comparison of the effects. In the second [4], the authors used a common thin film structure with electrical contacts on the sides, which is similar to the sample geometry used in our experiment. This comparison is presented in the Supplementary Information of the manuscript. We have clarified this in the main text. We have also modified our calculations by implementing the $\chi_{eff}^{(2)}$ characteristic.

We have added following estimations into the Supplementary Information:

Page 5:

“The effective second order susceptibility $\chi_{eff}^{(2)} = \frac{E_{2\Omega}}{E_{\Omega}^2} = \frac{39 \text{ V/cm}}{(148 \text{ kV/cm})^2} = 17.8 \text{ pm/V}$ ”

Also we modified sentence from “In comparison with large-area STEs, the peak power of optical excitation used there was about 8 GW/cm², while in our case it was only 58 MW/cm².” to “In comparison with large-area STEs, the peak power of optical excitation used there was about 8 GW/cm² (5.5mJ laser pulses, 40 fs duration, 4.8 cm beam diameter, generated THz pulse energy is about 5.1nJ), while in our case it was only 58 MW/cm².”

Page 6:

“The STE-based generated THz pulses (5.1nJ, 40 fs, 4.8 cm diameter) corresponds to 7 kW/cm².”

The conversion between peak power and the electric field can be done via $E = \sqrt{\frac{2 * \text{Peakpower}}{c * \epsilon_0}}$,

In this case the estimated $\chi_{eff}^{(2)}$ for STE is about 3.8pm/V

”

And

“The effective second order susceptibility $\chi_{eff}^{(2)} = \frac{E_{\Omega-\Omega}}{E_{\Omega}^2} = \frac{144 \text{ nV/cm}}{(60 \text{ V/cm})^2} = 0.4 \text{ pm/V}$ ”

5. From figure 1, it would appear that the ratio of spectral density between the carrier and the second harmonic, and the carrier and the rectified component is bigger than 10⁻⁷. This appears quite high, in view of the claimed efficiencies and requires a more detailed explanation.

Reply: In Figure 1, the fundamental power spectrum is multiplied by 10⁻⁷, which makes it almost the same amplitude as the TSHG and about twice smaller than the TOR signal. This means that the efficiency is greater than 10⁻⁷ for both TOR and TSHG signals. In the text on page 8 it was rewritten “The generated THz spectral densities are approximately 10⁻⁶-10⁻⁷ compared to the pump spectral density...”. We have corrected it to “The generated THz-wave spectral densities are more than 10⁻⁷ compared to the pump spectral density...”.

6. The Authors make the argument that the TSHG and TOR signals are detected only when the FM layer is sandwiched between layers of HM of opposite spin-Hall sign. The question arises if evidence can be provided in tri-layers rather than using bilayers?

Reply: In Figures 2c and 2d, it is evident that the TSHG and TOR signals are absent in trilayers with symmetric interfaces (Pt/Py/Pt, Ta/Py/Ta), while they are clearly visible in asymmetric trilayers such as Ta/Py/Pt. Additionally, Figure 2a provides evidence of TSHG in Py/Pt bilayers. The lack of TSHG signal in structures with symmetric interfaces is attributed to the out-of-phase emission of TSHG fields by the two symmetric interfaces, leading to destructive interference. The out-of-phase emission arises due to the opposite directions of the spin current in the two layers. However, when the interface symmetry is disrupted, TSHG can be readily detected. Trilayers with HMs of opposite signs of the spin Hall angle emit TSHG fields in phase, resulting in constructive interference. We use trilayers to demonstrate the coherent properties of the injected spin currents at the two distinct interfaces.

7. On Page 8 an explicit comment is made on the spectral densities of the generated THz waves, at electric field magnitude of 150 kV/cm. As the effects would be nonlinear both in the optical and THz regions, it begs the question as to what is the fair way of comparison – by intensity (squared amplitudes) or directly by intensity?

Reply: For a better comparison of the effects induced by laser and THz pulses we have introduced the $\chi_{eff}^{(2)}$ (please see also a reply to the first and fourth comments).

8. On the same page, bulk Pt conductivity is involved in the argument, only to be alleged for the discrepancy of TSHG field amplitude. It is quite clear that layers of such thicknesses cannot remotely reach bulk conductivities and a more refined estimates of conductivity may be needed (from optical complex refractive index measurements or perhaps microwave reflection/absorption measurements).

Reply: Indeed, the conductivity of a few nm thick metallic layers could be very different from that of bulk metals because of the strong contribution of interfacial/boundary scattering processes. The resistivity of the ultrathin Pt layers (from 2 nm to 20 nm thick) has been characterized in previous work [Ref. 35 in the main text], and its results are in line with our claims that metal resistivity strongly increases for lower film thicknesses. Here, the resistivity for 2 nm Pt layer is more than an order higher with respect to 20 nm film [35]. To crosscheck it, we performed additional resistance measurements for 2 nm and 10 nm thick Pt layers. The resistivity of 2 nm thick Pt layer was about 30 times higher than in 10 nm Pt layer. To clarify this in the text, we added following sentence on Page 9:

“We measured the resistivity of 2 and 10 nm thick Pt layers, and for the 2 nm layer it is about 30 times larger, which agrees well with previous work³⁵.”

Additionally, we performed resistance measurements on ultrathin 4 nm Pt layers at different temperatures. The temperature dependence of the Pt resistivity is shown in Supplementary Information Extended Figure 8. The resistivity increases monotonically with temperature, being about 1.44 times higher at room temperature than at helium temperatures. Such a change in resistivity should result in a twofold increase in TSHG at room temperature compared to TSHG at liquid helium temperature. The observed SHG enhancement is about 1.6 (Figure 4a), which is close to 2. This slight difference can be attributed to the different conductivity of 2 nm Pt grown on 2 nm Py with respect to 4 nm Pt layer. On manuscript page 7, we have added the following sentence:

“The temperature dependence of the resistivity of the 4 nm Pt layer is shown in the Supplementary Information, Extended Figure 8”.

9. The argument of Note 1 of the SI is very schematic. It appears that there is some dimensional factor – likely an elementary charge missing from equation 8.

Reply: Thank you. Yes, there is a missing factor of the elementary charge in Eq. 8. The expression 8 should read

$J_s = G \mu_B / e = \dots$

The expression 8 have been corrected in the Supplementary Information.

10. If the proportionality in Note 1 is based on the schematic square root, quasi-free electron gas dispersion in 3D, then it is probably better to have the final J_s to be expressed as a function of the Fermi level spin polarization and not be limited by an assumption of $\Delta = E_f / 2$.

Reply: The estimate is crude and e.g based on an approximate quadratic dispersion. While it is straight forward to give an expression as a function of the Fermi level spin polarization, we do not think such expressions significantly improves the estimate in metallic ferromagnetic where the exchange splitting is of the order of the Fermi energy.

11. Also in Note 1, the temporal aspect is lost after eq. 2. What assumptions are needed to justify this. The dimensionless factor R is not defined in eq. 1. And the Γ factors for electron-spin and electron-phonon coupling cannot possibly have dimension of time as of the way they enter the exponents.

Reply: We want to find expressions for the maximum spin accumulation before it decays. Therefore, we consider the instantaneous time $t=0$. The factor R described in Ref. 42 is then irrelevant. R is the ratio between the electronic and total heat capacity. Indeed, the Γ factors are rates and not times. We have corrected this in the revised Supplementary Information.

Reviewers' Comments:

Reviewer #1:

Remarks to the Author:

The authors have addressed all the comments raised in the review. The presentation of the experimental data and its interpretation is much improved in the revised version. The authors have considered a broader perspective and a more quantitative analysis of the observed dynamics is proposed. On this basis I am happy to recommend the manuscript for publication in Nature Communications.

1) Reviewer #1: The authors have addressed all the comments raised in the review. The presentation of the experimental data and its interpretation is much improved in the revised version. The authors have considered a broader perspective and a more quantitative analysis of the observed dynamics is proposed. On this basis I am happy to recommend the manuscript for publication in Nature Communications.

Response: We thank the reviewer for the positive assessment of our work.

Sincerely,

Sergey Kovalev, on behalf of all authors.